# Transforming a Large-Scale Prostate Cancer Outcomes Dataset to the OMOP Common Data Model—Experiences from a Scientific Data Holder’s Perspective

**DOI:** 10.3390/cancers16112069

**Published:** 2024-05-30

**Authors:** Nora Tabea Sibert, Johannes Soff, Sebastiano La Ferla, Maria Quaranta, Andreas Kremer, Christoph Kowalski

**Affiliations:** 1Health Services Research Department, German Cancer Society, 14057 Berlin, Germany; soff@krebsgesellschaft.de (J.S.); kowalski@krebsgesellschaft.de (C.K.); 2ITTM SA, Esch-sur-Alzette, 4354 Esch-sur-Alzette, Luxembourg; sebastiano.laferla@ittm-solutions.com (S.L.F.); maria.quaranta@ittm-solutions.com (M.Q.); andreas.kremer@ittm-solutions.com (A.K.)

**Keywords:** OMOP CDM, prostate cancer, observational dataset, data harmonisation

## Abstract

**Simple Summary:**

To obtain new insights in prostate cancer care, multiple clinics and research institutions must jointly analyse their data (by some, referred to as “big data analysis”). For this purpose, a common “data language” should be used. The OMOP CDM (common data model) is such a data language that can be used by different data holders. The aim of this article is to describe how a large dataset from a prostate cancer study with almost 50,000 prostate cancer cases was successfully transferred to the OMOP CDM standard. Challenges and their solutions during this process are discussed, and the importance of using data standards within prostate cancer research are reported from a scientific data holder’s perspective.

**Abstract:**

To enhance international and joint research collaborations in prostate cancer research, data from different sources should use a common data model (CDM) that enables researchers to share their analysis scripts and merge results. The OMOP CDM maintained by OHDSI is such a data model developed for a federated data analysis with partners from different institutions that want to jointly investigate research questions using clinical care data. The German Cancer Society as the scientific lead of the Prostate Cancer Outcomes (PCO) study gathers data from prostate cancer care including routine oncological care data and survey data (incl. patient-reported outcomes) and uses a common data specification (called OncoBox Research Prostate) for this purpose. To further enhance research collaborations outside the PCO study, the purpose of this article is to describe the process of transferring the PCO study data to the internationally well-established OMOP CDM. This process was carried out together with an IT company that specialised in supporting research institutions to transfer their data to OMOP CDM. Of n = 49,692 prostate cancer cases with 318 data fields each, n = 392 had to be excluded during the OMOPing process, and n = 247 of the data fields could be mapped to OMOP CDM. The resulting PostgreSQL database with OMOPed PCO study data is now ready to use within larger research collaborations such as the EU-funded EHDEN and OPTIMA consortium.

## 1. Introduction

Fortunately, more and more data from routine oncological health care, including i. a. information on diagnosis, treatment options, procedures, complications as well as data from (observational) longitudinal studies are electronically available. Whereas this shift to a digitalised data collection and data storage enables clinical researchers to have easier access to their own oncological data for analysis, the plethora of different data dictionaries (so-called vocabularies), data models, data formats and analysis tools often impedes multi-centre or international collaborations and research cooperations. Using the same vocabularies and common data models (CDMs) not only for a single study, but for routine clinical care data, may facilitate research and enhance replicable and transparent large-scale studies with different health data sources and from different health care providers [1,2,3].

Initiatives such as the International Consortium for Health Outcomes Measurement (ICHOM) [4], Core Outcome Measures for Effectiveness Trials (COMET) [5] or the COS-STAR [6] statement underscore the importance of using similar data collections from a data content’s perspective. Such core outcome sets define which outcomes are important for which condition and—in some cases—also which covariates should be collected. Core outcome sets are more and more often used in research [7,8]. However, not only *what* data should be collected, but *how* they should be collected regarding data content and structure can and should be harmonised. The Observational Medical Outcomes Partnership (OMOP) [9] administered by the Observational Health Data Sciences and Informatics (OHDSI) initiative offers a CDM especially designed for the standardisation of routine clinical data from different data sources [10]. Originally designed by a public–private partnership mandated by the U.S. Federal Drug Administration (FDA) in 2007, the OMOP CDM has since been used by many health care research initiatives aiming at collaboratively analysing data from different sources and nations [11,12,13].

In order to extend the OMOP CDM to data sources in Europe and make them accessible for federated analyses, the European Health Data & Evidence Network (EHDEN) was launched in 2018 [14]. It is an EU-funded platform of data partners and researchers aiming at using different data sources to answer research questions about routine clinical care that may need different sources other than a single study or study centre. The idea of the EHDEN network was taken up by the European Medicines Agency (EMA) and will also be continued within the framework of the DARWIN EU (deliver a sustainable platform to access and analyse health care data from across the EU) platform [15]. For prostate cancer, the idea of joint, cross-institutional data analyses has been adopted in Europe by the large multi-stakeholder consortium PIONEER [16,17]. Here too, data in the OMOP CDM standard are used to jointly answer questions about the routine care of prostate cancer internationally. Such large research initiatives are only possible if the researchers agree on a common “data language”. Such a data language is OMOP CDM, which is why the standard is also recommended for use by the EMA [15].

Since 2016, the German Cancer Society has been conducting the large Prostate Cancer Outcomes (PCO) study with, as of today, more than 63,000 patients enrolled by 148 prostate cancer centres from Germany, Switzerland and Austria. The PCO database consists of comprehensive information on diagnosis (i.e., Gleason scores, TNM, PSA), treatment options (i.e., Active Surveillance, radical prostatectomy, radiation), complications (CTCAE, margins), specialised care (i.e., nodes taken, psycho-oncological care, social counselling), follow-up information (i.e., biochemical recurrence, (disease-free) survival) as well as baseline and 12 months patient-reported outcomes (EPIC-26, [18]). The PCO study is part of the TrueNTH Global Registry (TNGR) [19] and its main goal is to compare outcomes after prostate cancer care across different health care providers [20,21]. Standardising prostate cancer research was a key part of the study from the beginning: the data collection (*what* is collected) for PCO and the TNGR is based on the ICHOM standard dataset for localised prostate cancer [22]. For prostate cancer centres taking part in the PCO study, a standardised data collection approach is used: prostate cancer centres firstly document the clinical care data within a tumour documentation system of their choice. Then, all prostate cancer centres use a data harmonisation tool (OncoBox) to transform their data to a shared data format based on an xml data standard used for certification purposes by the German Cancer Society. Those datasets are merged with patient survey data from the PCO study using the OncoBox Research tool (https://www.xml-oncobox.de/de/Zentren/ProstataZentren; accessed on 23 May 2024), leading to a large dataset combining clinical data from prostate cancer care and patient-reported survey data. Thus, the PCO study already uses a comprehensive data collection and harmonisation approach including almost 150 different health care providers. This approach has been successfully used for different research projects investigating routine prostate cancer care [7,9,10,11,12,13,14,15].

As a next step to enhance international and large-scale prostate cancer research, the German Cancer Society decided to use the information already within the PCO study harmonised data and transform it into the internationally well-established OMOP CDM, and by that method, be part of the EHDEN data partners. The aim of this paper is to describe this transformation process from the data holder’s perspective and to elaborate on the encountered facilitators and burdens during this process.

## 2. Materials and Methods

### 2.1. The OMOP CDM (Version 5.4)

In 2007, the U.S. FDA encouraged the initiation of the public–private OMOP that should enhance the use of observational, routine clinical care data (i.e., electronic health records, claims data) for a faster and more reliable evaluation of drugs and procedures approved (or to be approved) by the FDA (in terms of an active drugs surveillance system). The self-imposed goal of OMOP was thus to “develop the necessary technology and methods to refine the secondary use of observational data for maximizing the benefit and minimizing the risk of pharmaceuticals” [9]. A key part of the initiative from the very beginning was the development of a CDM for using different data sources in large-scale studies without the necessity to 1. centralise the datasets and to 2. have a unique data vocabulary and data infrastructure for each separate study and analysis. In 2012, Overhage et al. introduced the OMOP CDM, including a first validation on ten different datasets which aimed to harmonise both the content and structure of different observational data sources [23]. After the official end of the OMOP initiative, the OHDSI platform was built in 2014 as an open-sciences, multi-stakeholder collaborative to enhance the use of the OMOP CDM for large-scale “real-world evidence” research projects [24], and is up until now in charge of the maintenance and further development of the OMOP CDM. The OMOP CDM is updated regularly, with version 5.4 currently being the most up to date. Minor version changes (i.e., from 5.3 to 5.4) are backwards compatible, while major version changes may not be.

The OMOP CDM from a data holder’s perspective consists of three main features:A common, person-centric structured database framework based on a relational database approach with dedicated *data tables* (i.e., person table, measurement table, condition table, provider table, cf. Figure 1) to harmonise the data structure.A huge variety of *standardised vocabularies* (i.e., SNOMED, LOINC, etc.) which serve as a reference for all unique *concepts* that are mandatory to use within the OMOP CDM. This results in a comprehensive repository of *concept IDs* which are unique within the OMOP CDM and enable one to distinctly link information from the source data to an OMOP CDM concept. Concept IDs are specific to OMOP CDM, and codes from the integrated vocabularies are mapped to OMOP CDM concept IDs to ensure the uniqueness of the identifier. The online repository https://athena.ohdsi.org (accessed on 23 May 2024) offers an extensive overview of all concept IDs and vocabularies used for OMOP CDM.A *technologically independent* data model that is applicable to any relational database solution (i.e., SQL, Oracle, MariaDB).

For the below-described mapping to the OMOP CDM, version 5.4 was used.

### 2.2. The PCO Study Dataset

As of 2016, prostate cancer centres certified by the German Cancer Society can take part in the PCO study described above. The dataset—based on the ICHOM standard dataset [22]—consists of the following data field groups stored in an xml file, which, for analysis purposes, is transformed into a two-dimensional Excel workbook (selection of the most important data fields):Health care provider’s information: unique identifier, localisation (country, federal state);Patient’s sociodemographic information: unique identifier, age, citizenship (self-reported and documented by health care provider), insurance status (self-reported), highest school degree (self-reported), gender, date of death (if applicable);Patient’s case anamnesis: comorbidities (according to ICHOM standard dataset), previous cancer diagnosis;General prostate cancer case information: risk classification according to German S3-guideline [25], diagnosis date, cTNM, pTNM, Gleason scores at diagnosis and after surgery, PSA value at diagnosis and after surgery, ICD10 code, cores used for diagnosis (amount, percentages involved);Prostate cancer case treatment information: type of treatment (active surveillance, watchful waiting, radical prostatectomy, radiation, androgen deprivation treatment, other local and systemically treatment options), surgical approach (nerve-sparing, open/robotic/laparoscopic, lymphadenectomy incl. amount of taken nodes and amount of involved nodes), surgical revision procedures, date of treatment procedures, complications (Clavien–Dindo, CTCAE), margins, fraction means and amount of radiation;Prostate cancer case procedures’ information: tumour board dates, psycho-oncological counselling, social counselling;Prostate cancer case follow-up information: biochemical recurrence, number and years of follow-up, vital status, local recurrence and other malignancies after prostate cancer;Patient-reported outcomes: EPIC-26 scores and single items before and one year after diagnosis [18].

For the purposes of this project, the dataset was transferred from the original xlsx data format to a csv file.

### 2.3. German Cancer Society’s Data Transformation Approach

As the EHDEN data partner grant recipient, the German Cancer Society committed itself to finish the transformation to the OMOP CDM within one year with the support of a SME (small to medium enterprise). ITTM as an IT company with extensive knowledge through the mapping of several different data sources was chosen as the SME collaborator for the mapping and setting-up of the technical infrastructure. The following approach and technical infrastructure were chosen to map the PCO study dataset to the OMOP CDM (the several steps are described in detail below):Conceptual **vocabulary mapping** of each applicable source data field to an OMOP CDM concept and data table(s) using OMOP vocabularies;Developing and testing an efficient ETL (extract-transform-load) process based on a mock dataset within a VM (virtual machine) environment (**ETL development and testing**);Setting up of a **production VM** that hosts the final PostgreSQL database with data using the OMOP CDM format.

#### 2.3.1. Vocabulary Mapping

As a first step, the OHDSI-provided tool “White Rabbit” [26] was used for an initial source data scan. Based on this scan, for each data field of the PCO study dataset, the following was decided: Should the data field be mapped to the OMOP CDM? If yes: For which table of the OMOP CDM should the information of the data field be used? Which concept IDs from the OMOP CDM should the information be mapped to, and which rules do apply for those mappings?

The main resource for mapping decisions was Athena and the OHDSI-provided “Usagi” tool [27]. All unclear mappings were discussed by the project team members from the German Cancer Society and ITTM. If several mappings were feasible, either the more commonly used OMOP CDM vocabulary was used, or the scientific lead from the German Cancer Society (N.T.S) decided based on their health services research perspective’s expertise and ITTM’s former mapping experience. For any remaining concepts that were not represented by Athena, a custom vocabulary (“DKG_OMOP”) was created (further information upon request).

#### 2.3.2. ETL Development and Testing

The Pentaho Data Integration Community Edition (PDI-CE) version 9.4 software was used for ETL development. Based on a synthetic dataset generated and provided by the German Cancer Society team and the vocabulary mappings, ITTM developed a first ETL process. Within the Pentaho framework, a sequence of so-called jobs and transformations is specified, where 1. the PostgreSQL database is initialised including the setting up of necessary schemas and populating the tables with the vocabulary mapping information and source data, then 2. the source data are mapped to the OMOP CDM tables based on the vocabulary mappings (transformations) and finally 3. tables’ constraints are re-built where needed.

After the initial development of the ETL jobs and transformation scripts by ITTM, these were transferred to the ETL VM only accessible to the German Cancer Society team members (N.T.S., J.S.). Using the original PCO dataset, the ETL procedures were iteratively tested, and debugging was performed where needed jointly by ITTM and the German Cancer Society. Test queries on the source as well as on the OMOPed dataset were applied to ensure the accuracy and correctness of the mapping, as well as descriptive summary statistics performed in R. Additionally, the EHDEN-provided package “CDMInspection” including the DataQuality dashboard [28] was used for external quality control.

#### 2.3.3. Production VM with Final PostgreSQL Database

After the successful completion of the ETL process, the PCO study dataset was transformed into OMOP CDM on the ETL VM. The PostgreSQL containing only the OMOPed database was then transferred to the final production VM. OHDSI tools such as ATLAS version 2.15 [29] and HADES version 1.13 [30] were installed on the production VM in order to enable future collaborations, i.e., within the EHDEN and OPTIMA consortia and connected to the PostgreSQL database. The production VM was run on an Intel(R) Xeon(R) Platinum 8268 server with the following technical specifications: CPU 2.90GHz, 8 cores, RAM 16 GiB, storage 80 GiB.

## 3. Results

### 3.1. Source Dataset Specific Characteristics

For this data transforming, the last available PCO study dataset from 2023 with a total of N = 49,692 prostate cancer cases and 318 data fields was used. The oldest prostate cancer diagnosis date was 18 June 1997, and the latest was from 2 March 2023. For the PCO study, it is technically possible to include patients with more than one case. However, to improving the consistency of the dataset, it was decided by the scientific lead of the German Cancer Society (N.T.S) to exclude patients with more than one case recorded (n = 392/0.8% cases excluded). This resulted in a dataset in OMOP CDM with only newly diagnosed prostate cancer, and where every patient equalled one prostate cancer case.

### 3.2. ETL Process

The following tables of the OMOP CDM were populated with information from the PCO study: localisation, care_site, person, death, observation_period, visit_occurence, procedure_occurence, condition_occurence, measurement, observation and episode.

Vocabulary mapping was performed for n = 247 data fields. Exclusion of the other data fields was based on the decision to exclude data fields that were specific only to the PCO study’s infrastructure or certification requirements from the German Cancer Society (e.g., data fields in which content was present in multiple data fields or was calculated based on data fields that should be mapped); five data fields were used for several mappings. Table 1 shows the total number of records and the number of distinct persons with records in that table. On average (median), a person was mapped to 15 observations, 15 measurements, 14 condition occurrences and 5 procedure occurrences. In Appendix A, the 25 most-frequent concepts for condition, observation, measurement and procedure can be found.

To better illustrate the mapping process, please refer to Table 2 and Table 3 as two examples for the mapping of firstly “Surgery visit” (visit_occurence table) and secondly “Pre-treatment UrinaryFunction score” (observation table).

For “surgery visit” (Table 2), the mapping was straightforward. If the data field “SurgeryDate” of the PCO source dataset was filled (mapping condition), the following occurred:-A new and unique ID for the visit is generated (visit_occurence_id);-The kind of visit is set to “inpatient visit” (conceptID: 9201, visit_concept_ID);-The visit_start_date and visit_end_date are both set on the same date (SurgeryDate), since no end date of the surgery is available in the PCO dataset and surgery is most likely finished on the same day as it started;-Provenance of visit entry is set to “EHR” (conceptID 32187, visit_type_concept_id);-The “care_site_id” is filled with the identifier of the care_site associated with the value of the source field “RegNr” (pseudonym of health care providers within the source dataset) and the fieldname “SurgeryDate” is copied to visit_source_value.

All other fields were either not mandatory or not applicable.

For “Pre-treatment UrinaryFunction score” (Table 3), which relies on the EPIC-26 score “Urinary Function”—a patient-reported outcome score specific to prostate cancer [18,31]—the mapping was more complicated, because up until now, there was no consistent way of mapping patient-reported outcome data within the OMOP CDM framework. Thus, the decision was made to use concepts from the standardized vocabularies to capture the clinical meaning of the patient-reported outcome field as closely as possible. Consequently, the standard concept “Assessment score” was used as observation_concept_id, storing the numeric score in the field value_as_number. The context of the score was specified by the qualifier concept “Urinary system function”, while the concept “Survey” was used as observation_type_concept_id, to indicate the provenance of the data. The remaining mapping was equivalent to the above-mentioned. This can be compared with Appendix A to obtain an overview of all used vocabularies.

### 3.3. Quality Assurance

Table 4 shows selected summary statistics for the dataset transferred to OMOP CDM based on the dashboard information provided by the ATLAS tool [29], as well as a comparison to the results from the same queries on the source dataset.

## 4. Discussion

The PCO study dataset is a large-scale dataset combining data from routine prostate cancer care from more than 140 different health care providers with survey data on patient-reported outcomes and sociodemographic information. The aim of the described project was to transform this dataset to an internationally well-established, open-source data model to enhance research collaborations and improve transparency and replicability of health services research projects based on the PCO study dataset. Only 0.8% of prostate cancer cases from the source dataset were excluded beforehand and thus were not mapped; 78% of data fields were eligible to be mapped to an OMOP CDM table.

Other clinical registries reported much lower percentages of successful mapping. Biedermann et al. for instance reported, for their mapping of three drug registries on pulmonal hypertension, between 7 and 52% of non-mapped records [32], which is explained by a lack of information and thus the impossibility to populate mandatory OMOP CDM tables. The extremely high coverage and transfer of the PCO source data information to the dataset in the OMOP CDM format underscores that the OMOP CDM framework is well-suited for datasets that are based on clinical care information systems. This is supported by other examples from oncological care as well that reported comparable coverage of their source dataset compared to their OMOPed database [33].

Most challenges during the ETL process arose while mapping survey and especially patient-reported outcome data. Up until now (version 5.4), the OMOP CDM did not include a consistent and straightforward approach to map survey and patient-reported outcome data, a problem that has been discussed by other research groups already [34]. This is a disadvantage for initiatives like the PCO study group because important information of study administration such as an online or a paper and pencil approach of the survey become lost. From a health services research perspective, this information is highly important since researchers can understand the context of data collection a lot better if information is available in the CDM [35]. For the purpose of this first mapping, three simultaneous approaches to dealing with patient-reported outcome data were applied:

Firstly, the patient-reported outcome scores (as “PreUrinaryFunction”) were mapped to Observation with “Assessment score” as observation_concept_id, storing the numeric score in value_as_number and using the concept “Survey” as observation_type_concept_id.

Secondly, the items from the EPIC-26 (such as “Overall, how big a problem have your bowel habits been for you during the last 4 weeks?”) were mapped to Condition_occurrence and/or Observation with concepts that captured the clinical meaning of the answer. In observation, based on the answer, the fields observation_concept_id, value_as_concept_id and qualifier_concept_id were filled with appropriate concepts to represent the severity degree of a symptom or the level of frequency of a certain behaviour. In fact, the severity degree and the frequency level of the symptom/behaviour were reported using, respectively, the concepts “Condition severity” and “Frequency of behaviour”. As an example, for item no. 7 from the EPIC-26 (“Overall, how big a problem have your bowel habits been for you during the last 4 weeks?”), a Condition_occurrence record was created with either the concept “Irregular bowel habits” (ID: 40480640, from SNOMED) if the answer of the patients was not “no problem”, or the concept “Normal bowel habits” (ID: 4208260, from SNOMED) for “no problem”. At the same time, the severity of the problem, depending on the answer (“Very small problem”, “Small problem”, “Moderate problem”, “Big problem”), was mapped to Observation and linked to the Condition_occurrence record.

Thirdly, whenever it was not possible to find concepts that adequately reflected the source meaning, custom concepts were adopted. Although those three approaches allow for representing the information as conditions/clinical findings within the OMOP CDM and partly as survey-based, important information about the provenance of the findings (a patient-reported outcome questionnaire) becomes lost.

Therefore, the health services research team from the German Cancer Society already actively got in contact with OHDSI as the maintaining organisation of the OMOP CDM and worked on a comprehensive way to include patient-reported outcomes within the framework without losing too much meta-information. This would be a major advantage for further developing the OMOP CDM and increasing the usability for patient-centred clinical research. Especially for oncological conditions with long disease-free survival such as prostate cancer, symptoms and functions that are directly reported by patients are highly relevant and more and more key outcomes for any prostate cancer research projects [20,36,37]. Thus, we encourage all OHDSI collaborators and researchers interested in using routine oncological care data for scientific purposes to find solutions to incorporate patient-reported outcomes to the OMOP CDM.

Another issue that arose due to the specific nature of the source dataset was the handling of drug exposures for which the specific pharmaceutical agent was not documented, but the type of treatment was. For instance, the PCO dataset did include information about whether an androgen deprivation therapy was applied to the patient or not. However, the dataset did not include information about the specific drug that was prescribed. Thus, this information was not populating the drug_exposure table, but rather the more general procedure_occurence table.

The described data transfer process was not only a major milestone for the PCO study group or the internationally collaborating TNGR initiative. It was also, to the authors’ knowledge, the first project describing how the ICHOM standard dataset for localised prostate cancer (cf. introduction for more details)—defining what should be collected in the sense of a core/standard outcomes set—is mapped to a CDM, defining how the data should be collected and stored. This process was successful, and we would encourage other ICHOM standard dataset users to map their data to the OMOP CDM as well, since the combination of core outcome sets and CDMs is an important next step to enhance international collaborative research not only in urological oncology.

This data transfer project was supported by two important pillars that led to the swift and successful completion of the project: the German Cancer Society, on the one hand, as a data holder and institution with considerable data expertise, and on the other hand, ITTM as a specialised company that supported the scientific working group in the technical implementation. Only through such a combination of scientific expertise, data holders, IT expertise and knowledge of the OMOP CDM, is it possible to create a sustainable data infrastructure that will enable future research projects based on the OMOP CDM standard. The collaboration between the German Cancer Society and ITTM included extensive ETL training offered by ITTM to the scientific team in order to enable the German Cancer Society to adapt the described ETL process for future possible changes to the dataset (or the OMOP CDM version). By this, a frictionless re-run of the ETL—planned at least annually for the dataset update—is assured.

A key feature of using the OMOP CDM is the possibility to analyse different datasets using the same analysis protocols and scripts without the necessity to merge datasets centrally beforehand (the so-called “federated data analysis” methodology for such a project within PIONEER, i.e., described by [38] or [39]). Federated data analysis approaches are strongly supported not only by the scientific community but also from a legislative perspective including inter alia the European Commission with their view on the European Health Data Space [40]. This is why platforms such as DARWIN EU, EHDEN and ongoing consortia such as the OPTIMA consortium encourage data holders to join their initiatives. The German Cancer Society is part of some of those initiatives. By transferring the PCO dataset to the OMOP CDM, the German Cancer Society hopes to enhance prostate cancer care research internationally. The sustainability of the newly built research platform was and is guaranteed during the project by receiving IT training from ITTM for updating the OMOP CDM version. This means that new OMOP concepts (e.g., for patient-reported outcomes) can be integrated by updating the OMOP CDM version, either alone or in combination with a data update (e.g., to a new PCO dataset).

Although those federated data analyses allow much more extensive insights into clinical care from many different perspectives, care settings, countries and caregivers, information about the differences between the datasets included in such an analysis and their individual selection biases and restrictions may get lost. This highlights the importance of well-documented and freely available meta-data on each dataset that must be accessible to potential research collaborators. The EHDEN portal (https://portal.ehden.eu/summary, accessed on 23 May 2024) gives a first overview of those meta-data by the required fingerprinting forms that each data partner must fill out. However, special information on the unique particularities of the datasets that may be taken into account during the analysis are often not shared publicly, and thus, possible limitations of the federated data analysis may not be as evident as for single-dataset studies. We therefore encourage all data holders who transfer their data to the OMOP CDM and are interested in federated data analyses to be as transparent as possible about the peculiarities of their datasets.

## 5. Conclusions

Transforming a large, well-established prostate cancer dataset from more than 140 different health care providers to the OMOP CDM is feasible. Despite some data fields that were not meaningful outside the PCO study and were thus not planned to be mapped, for a high amount of the 78% of the information captured by the source dataset, a fitting OMOP CDM table and/or concept could be found. Using an internationally well-established data model enables researchers to further collaborate. For the OMOPed PCO study dataset, this next step is already being performed since the German Cancer Society takes part in, for instance, the European-wide OPTIMA and EHDEN consortia.

## Figures and Tables

**Figure 1 cancers-16-02069-f001:**
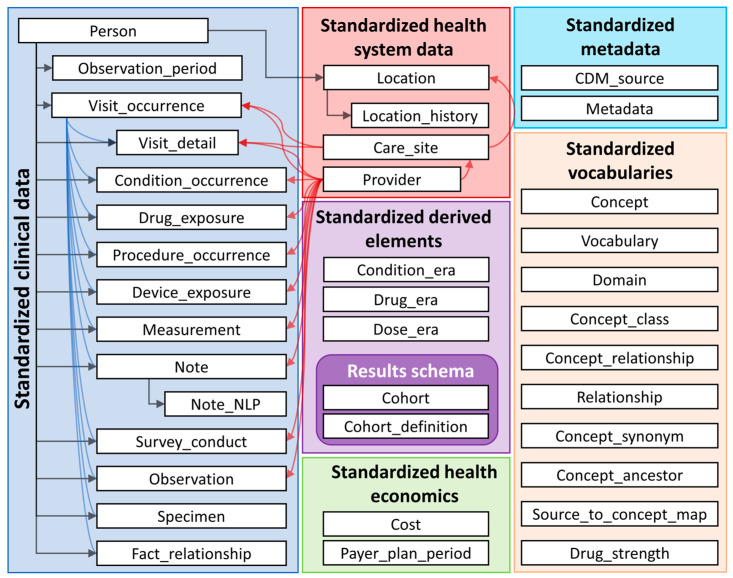
Data tables from the OMOP CDM [10].

**Table 1 cancers-16-02069-t001:** Number of records and persons per OMOP CDM table.

Tablename from OMOP CDM	Count	Number of Persons
observation	1,507,707	49,300
condition_occurrence	862,669	49,286
measurement	801,580	49,300
procedure_occurrence	227,540	49,300
observation_period	175,675	49,300
visit_occurrence	65,098	48,358
person	49,300	49,300
death	721	721
care_site	148	NA
location	38	NA
provider	0	NA
specimen	0	0
dose_era	0	0
device_exposure	0	0
visit_detail	0	0
drug_era	0	0
condition_era	0	0
cost	0	NA
note	0	0
drug_exposure	0	0
payer_plan_period	0	0

**Table 2 cancers-16-02069-t002:** The visit_occurence table for “Surgery visit”.

Topic	Definition
period	Diagnosis, pre-treatment PCO survey, treatment period
mapping condition	SurgeryDate is not null or empty
visit_occurrence_id	Generate a unique visit_occurrence_id
person_id	Find the person_id in the person table where person_source_value = PatientID
visit_concept_id	Use the concept_id 9201 “Inpatient Visit”
visit_start_date	Copy paste the date SurgeryDate
visit_start_datetime	
visit_end_date	Copy paste the date SurgeryDate
visit_end_datetime	
visit_type_concept_id	Use the concept_id 32817 “EHR”
provider_id	
care_site_id	Find the care_site_id in the care_site table where case_site_source_value is the same as the current value of RegNr
visit_source_value	Copy paste the source fieldname “SurgeryDate”
visit_source_concept_id	
admitted_from_concept_id	
admitted_from_value	
discharge_to_concept_id	
discharge_to_source_value	
preceding_visit_occurence_id	

**Table 3 cancers-16-02069-t003:** Observation table for “Pre-treatment UrinaryFunction score”.

Topic	Definition
period	Diagnosis, pre-treatment PCO survey, treatment period
mapping condition	PreUrinaryFunction is not null or empty and PreDate is not null or empty
observation_id	Generate a unique observation_id
person_id	Find the person_id in the person table where person_source_value = PatientID
observation_concept_id	Use concept_id 36684305 “Assessment score”
observation_date	Copy paste the date PreDate
observation_datetime	
observation_type_concept_id	Use 32883 “Survey”
value_as_number	Copy paste the value of PreUrinaryFunction
value_as_string	
value_as_concept_id	
qualifier_concept_id	Use the concept_id associated to the source fieldname “PreUrinaryFunction”
unit_concept_id	
provider_id	
visit_occurence_id	
visit_detail_id	
observation_source_value	Copy paste the fieldname PreUrinaryFunction
observation_source_concept_id	
unit_source_value	
qualifier_source_value	
value_source_value	Copy paste the fieldname PreUrinaryFunction
observation_event_id	
obs_event_field_concept_id	

**Table 4 cancers-16-02069-t004:** Selected quality assurance checks (descriptive summary statistics comparison between source data and the OMOP CDM transformed dataset).

Characteristic	Result Source Dataset (with Multiple Cases Excluded)	Result OMOPed Dataset	Reason for Discrepancy
number of patients with cM0	49,300	49,300	-
number of patients with pM0	42,925	42,875	Additional mapping rule: SurgeryDate must be unequal 0
number of patients with Clavien–Dindo Grade I	1931	1931	-
number of patients with lymphadenectomy	38,036	38,036	-
minimum of year of birth	na	1932	Birth year is not a data field of the source dataset, but is calculated for the OMOPed dataset by subtracting the age at diagnosis from year of diagnosis

## Data Availability

Due to GDPR requirements and the contracts with the participating study centres, we are not able to share individual patients’ data obtained for the PCO study purposes. This is in accordance with the Ethics Committee of the Medical Association of Berlin which can be contacted and requests for data placed via ed.bkea@ke.

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
