# Peer review of "Transforming a Large-Scale Prostate Cancer Outcomes Dataset to the OMOP Common Data Model—Experiences from a Scientific Data Holder’s Perspective"

_cancers, 2024, doi:10.3390/cancers16112069_

Round 1

Reviewer 1 Report

Comments and Suggestions for Authors

I would like to thank the authors for this manuscript and their great effort. The manuscript is well written and clearly points out the advantage of harmonizing data.

There are a few comments.

- The PRO data is being converted to OMOP which contains of data from 143 centers (lines 73-74). Does this data includes all national data?

- In table 1, the counts of "drugs" is zero. How was the initiation of ADT recorded in the data and how is this reported in the OMOP'ed data? 

- In table 1, there were 148 care sites. However, in line 74 it is mentioned that the PCO study contains data of 143 prostate cancer centers. This seems a discrepancy.

- Some information in the PCO database does not have a standard concept in Athena (like distant-free survival). How was this handled? I am not sure whether the authors developed a custom vocabulary for those variables.

Minor comments

- Line 104&117. OMOP and OHDSI are already written in full in the Introduction so it should be abbreviated. 

- The authors describe their ETL. Are the authors planning to regularly update their data?

Sebastiaan Remmers

Author Response

Thank you very much for the time and effort to review our manuscript and your valuable comments. Please find below our responses in italic (quotes from the revised in manuscript are normally set):

There are a few comments.

- The PRO data is being converted to OMOP which contains of data from 143 centers (lines 73-74). Does this data includes all national data?

Thank you verry much for the question. No, it is not mandatory for prostate cancer centres to be firstly certified by the German Cancer Society and secondly, to take part in Prostate Cancer Outcomes Study (PCO study) – the base of PRO data.

- In table 1, the counts of "drugs" is zero. How was the initiation of ADT recorded in the data and how is this reported in the OMOP'ed data? 

Thank you very much for the possibly to clarify this issue. Unfortunately, for some pharmaceutical intervention, the data contains information about the type of treatment (i.e. ADT) but not the specific drug that was given. Hence, it was not possible to map to a specific drug concept, we included the information in visit_occurence and procedure_occurence tables. Since we consider this not only a specific problem to our dataset, we added the following to the discussion: “Another issue that arose due to the specific nature of the source dataset was the handling of drug exposures for which the specific pharmaceutical agent is not docu-mented but the type of treatment: For instance, the PCO dataset does include infor-mation whether an androgen deprivation therapy was applied to the patient or not. However, the dataset does not include information about the specific drug that was prescribed. Thus, this information is not populating the drug_exposure table but rather the more general procedure_occurence table.”  

- In table 1, there were 148 care sites. However, in line 74 it is mentioned that the PCO study contains data of 143 prostate cancer centers. This seems a discrepancy.

Thanks for your attentive reading, 143 was indeed a typo and we corrected the text accordingly.

- Some information in the PCO database does not have a standard concept in Athena (like distant-free survival). How was this handled? I am not sure whether the authors developed a custom vocabulary for those variables.

Your are absolutely right: For all information for which we did not find a standard concept in Athena, a custom vocabulary was created (“DKG_OMOP”), we added the following to the section Material and Methods: “For any remaining concepts, that were not represented by Athena, a custom vocabulary (“DKG_OMOP”) was created (further information upon request).

Minor comments

- Line 104&117. OMOP and OHDSI are already written in full in the Introduction so it should be abbreviated. 

Thanks, we corrected the issue.

- The authors describe their ETL. Are the authors planning to regularly update their data?

Yes, thanks for the opportunity to clarify the issue, we added to the discussion: “The collaboration between German Cancer Society and ITTM included an extensive ETL training offered by ITTM to the scientific team in order to enable the German Cancer Society to adapt the described ETL process for future possible changes to the dataset (or the OMOP CDM version). By this, a frictionless re-run of the ETL – planned at least annually for the dataset update – is assured.

Reviewer 2 Report

Comments and Suggestions for Authors

The authors have described the process of OMOPing data from a large data provider, i.e. the German Cancer Society, into a dataset that can be used in big data consortia and other collaborative research projects. Although my understanding of the process of OMOPing data is limited, I have a few questions:

*Simple summary, line 10: would the authors like to mention here the concept of big data? Because that is what you are referring to, when you say that multiple clinics and research institutions have to jointly analyze their data? Please adjust ‘jointly analysis (=analyze) their data’.
*Simple summary, line 12: perhaps it is wise to write CDM the first time it is used?
*Simple summary, line 13: ‘The aim of this study is to describe (instead of described)..’.

*Introduction, lines 73-82: the PCO study database includes comprehensive information on different aspects of the prostate cancer care path; i.e. diagnosis, treatment, follow-up. How does this relate to databases from other data sources? From big data projects we have learned that claims database for instance, does not have such comprehensive information available.

*Materials and methods, the OMOP CDM: do data have to be in the same OMOP CDM version to be compatible?
* Results, line 243: ‘with more than one (instead of once) case recorded’.

*Discussion, lines 348-356, example of the EPIC (bowel habits): are you still able to use the scripts from the EPIC to calculate domain scores when EPIC-26 data is OMOPed? Or do you also need to OMOP the domain scores?

*Discussion, lines 383-384: the German Cancer Society and ITTM have collaborated closely in this project. How does it work if new data becomes available? Are yearly updates of German Cancer Society database OMOPed for instance? Will data providers such as the German Cancer Society always need to rely on an IT-partner, such as ITTM, to OMOP their data? Or is it a process, once well-established for a certain dataset, that can be completed by data providers themselves too? Does this influence the future perspective of OMOPing data, the costs and opportunities that come with it?

Comments on the Quality of English Language

There are some small spelling errors here and there. 

Author Response

Thank you very much for your time and effort to review our manuscript. Please find below our comments in italic with quotes from the revised manuscript set normally!

*Simple summary, line 10: would the authors like to mention here the concept of big data? Because that is what you are referring to, when you say that multiple clinics and research institutions have to jointly analyze their data? Please adjust ‘jointly analysis (=analyze) their data’.

Thank you very much. The concept “big data” is sometimes miss-leading which is the reason why we avoided to use it within the article. However, you are right that exp. For the simply summary, mentioning the concept may help some readers to get a first impression about the main content of the article. We therefore changed accordingly: “To get new insights in prostate cancer care, multiple clinics and research institutions have to jointly analyse their data (by some referred to as “big data analysis”).
*Simple summary, line 12: perhaps it is wise to write CDM the first time it is used?

Thanks, we added the information.
*Simple summary, line 13: ‘The aim of this study is to describe (instead of described)..’.

Thanks, we corrected the typo!

*Introduction, lines 73-82: the PCO study database includes comprehensive information on different aspects of the prostate cancer care path; i.e. diagnosis, treatment, follow-up. How does this relate to databases from other data sources? From big data projects we have learned that claims database for instance, does not have such comprehensive information available.

Thank you very much for the opportunity to clarify the nature of the PCO dataset: it includes information that is mostly comparable with cancer registry data with additionally having information directly collected by patients’ surveys (such as socio-demographic background, patient-reported outcomes, etc.). The quality of the data is guaranteed since it is audited each year during (re-) certification process of the centres.

*Materials and methods, the OMOP CDM: do data have to be in the same OMOP CDM version to be compatible?

Thanks for the question, the version combability depends as described and added to the Materials and Methods section: “The OMOP CDM is updated regularly, with version 5.4 currently being the most up-to-date. Minor version changes (i. e. from 5.3 to 5.4) are backwards compatible, major version changes may not.”
* Results, line 243: ‘with more than one (instead of once) case recorded’.

Thanks, we corrected the spelling error!

*Discussion, lines 348-356, example of the EPIC (bowel habits): are you still able to use the scripts from the EPIC to calculate domain scores when EPIC-26 data is OMOPed? Or do you also need to OMOP the domain scores?

Thanks for that very good question: In order to not loose information of the EPIC-26 domain scores, we mapped both the item scores as well as the five EPIC-26 domain scores. For future initiatives, we would strongly support a system that enables to map PROs single items without domain scores and save domain calculation scripts as meta information together with the type of questionnaire.

*Discussion, lines 383-384: the German Cancer Society and ITTM have collaborated closely in this project. How does it work if new data becomes available? Are yearly updates of German Cancer Society database OMOPed for instance? Will data providers such as the German Cancer Society always need to rely on an IT-partner, such as ITTM, to OMOP their data? Or is it a process, once well-established for a certain dataset, that can be completed by data providers themselves too? Does this influence the future perspective of OMOPing data, the costs and opportunities that come with it?

Thanks for those highly relevant questions, we added the following to the discussion to describe the (sustainability of the) collaboration between ITTM and the German Cancer Society better (discussion): “The collaboration between German Cancer Society and ITTM included an extensive ETL training offered by ITTM to the scientific team in order to enable the German Cancer Society to adapt the described ETL process for future possible changes to the dataset (or the OMOP CDM version). By this, a frictionless re-run of the ETL – planned at least annually for the dataset update – is assured.

Reviewer 3 Report

Comments and Suggestions for Authors

No suggestions